# Construction of Dimeric Drug-Loaded Polymeric Micelles with High Loading Efficiency for Cancer Therapy

**DOI:** 10.3390/ijms20081961

**Published:** 2019-04-22

**Authors:** Bing Yu, Qingye Meng, Hao Hu, Tao Xu, Youqing Shen, Hailin Cong

**Affiliations:** 1Institute of Biomedical Materials and Engineering, College of Chemistry and Chemical Engineering, College of Materials Science and Engineering, Qingdao University, Qingdao 266071, China; yubingqdu@163.com (B.Y.); mengqingye94@163.com (Q.M.); huhao@qdu.edu.cn (H.H.); 15165520337@126.com (T.X.); amnano@163.com (Y.S.); 2Laboratory for New Fiber Materials and Modern Textile, Growing Base for State Key Laboratory, Qingdao University, Qingdao 266071, China; 3Key Laboratory of Biomass Chemical Engineering of Ministry of Education, Center for Bionanoengineering, and Department of Chemical and Biological Engineering, Zhejiang University, Hangzhou 310027, China

**Keywords:** dimeric drug, camptothecin, micelles, cancer therapy

## Abstract

Polymeric micelles (PMs) have been applied widely to transport hydrophobic drugs to tumor sites for cancer treatment. However, the low load efficiency of the drug in the PMs significantly reduces the therapeutic efficiency. We report here that disulfide-linked camptothecin (CPT) as a kind of dimeric drug can be effectively embedded in the core of poly(ε-caprolactone)–poly(ethylene glycol)–poly(ε-caprolactone) (PCL–PEG–PCL) PMs for improving drug-loading efficiency, and PEG can be used as a hydrophilic shell. Moreover, the dimeric CPT-loaded PCL–PEG–PCL PMs exhibited excellent solubility in phosphate-buffered saline (PBS) media and significant cytotoxicity to cancer cells.

## 1. Introduction

Hydrophobic anticancer drugs, such as doxorubicin, camptothecin and paclitaxel have attracted great attention for treating all sorts of malignant tumors [1,2,3]. However, they have suffered from many limitations during the process of treatment, such as side effects, non-specific immunity of humans, and aggregation in vitro and in vivo [4,5]. In order to solve the above problems, researchers used nanocarriers to deliver anticancer drugs. As a result, polymeric micelles have been widely investigated for drug delivery because they show high stability and extended circulation in the blood [6,7,8]. In particular, poly(ethylene glycol) (PEG)-functionalized polymeric micelles exhibited favorable biocompatibility and can effectively inhibit nonspecific absorption to blood plasma proteins and increase stability [9,10,11].

Based on the above reasons, we developed a biocompatibility and biodegradability polymeric micelles based on poly(ε-caprolactone)–poly(ethylene glycol)–poly(ε-caprolactone) (PCL–PEG–PCL) triblock copolymer for efficient delivery of CPT. In order to improve the load efficiency, a dimeric camptothecin derivative (CPT–SS–CPT) was prepared by a reaction of cystamine with CPT. The resulting CPT–SS–CPT can prevent prodrug aggregation or crystallization into large aggregates. Thus, self-assembly of the CPT–SS–CPT and PCL–PEG–PCL with high load efficiency were constructed by hydrophobic interaction. Subsequently, the morphological structure, size, solubility, loading efficiency and in vitro cytotoxicity were assayed.

## 2. Results

Amphipathic PCL–PEG–PCL triblock copolymer as the ideal drug carrier was readily synthesized via the ring-opening polymerization of CL with PEG as a macroinitiator, and Sn(Oct)2 was added as a catalyst, as shown in Figure 1a.

The drug-loading efficiency and drug-loading content of CPT–SS–CPT/polymeric micelles (PMs) were calculated by high-performance liquid chromatography (HPLC) (Figure 2). The Table 1 indicates that the drug-loading efficiency and drug-loading content of CPT–SS–CPT/PMs is six times higher than CPT-loaded PCL–PEG–PCL PMs. As shown in Figure 3a, a transmission electron microscope (TEM) image showed the spherical morphology, and the particle size (approximately 40–60 nm) of CPT–SS–CPT/PMs is in accordance with a result measured by DLS (Figure 3b). Figure 4 presented the absorbance spectra of the dimeric CPT in solvent and the CPT–SS–CPT-loaded PCL–PEG–PCL PMs in aqueous solution. The absorption peaks of CPT–SS–CPT and CPT–SS–CPT/PMs center maintained at approximately 210 nm, 280 nm and 370 nm without distinct shifts.

The stability of CPT-loaded PCL–PEG–PCL PMs and CPT–SS–CPT/PMs was evaluated using photographs. Figure 5a exhibited the photograph of CPT-loaded PCL–PEG–PCL PMs dispersed in phosphate-buffered saline (PBS). The precipitation phenomenon of CPT was observed from PMs after 4 h of incubation. However, CPT–SS–CPT/PMs dispersions are maintained stable for 4 h with no obvious the precipitation (Figure 5b). X-ray diffraction (XRD) patterns of raw CPT and CPT–SS–CPT are shown in Figure 6 and clearly indicate that the serval characteristic crystalline peaks of the CPT–SS–CPT in the XRD diffractogram at the diffraction angles of 2θ = 9.0°, 11.9°, 13.3°, 17.7°, 24.9° and a broad peak (30°–50°) were completely and almost disappeared in contrast to the XRD diffractogram of CPT.

The cell viability of the modified CPT and CPT-loaded micelles at various concentrations was evaluated in HepG2 and NIH3T3 cells by using a MTT assay (Figure 7a). As shown in Figure 7b, it was observed distinctly that HepG2 cells were inhibited or killed when treated with CPT–SS–CPT/PMs at 0.5 μL/mL. The cells treated with drugs with disulfide bonds (CPT–SS–CPT) demonstrated significant amount of red fluorescent cells.

## 3. Discussion

In order to improve load efficiency and inhibit drug aggregation, the dimeric camptothecin (CPT–SS–CPT) was synthesized by activating the two end amino groups of cystamine with DIAE followed by reaction with an excess 4-Dimethylaminopyridine (DMAP) activated triphosgene. The CPT–SS–CPT-loaded PCL–PEG–PCL PMs were constructed by self-assembly of CPT–SS–CPT and the triblock copolymer using a solvent evaporation strategy (Figure 1b) [12]. The CPT–SS–CPT was effectively encapsulated in polymeric micelles by hydrophobic interaction between the hydrophobic segments in PMs and the CPT–SS–CPT. This interaction can facilitate self-assembly to the formation of micelles with a hydrophobic drug core and a hydrophilic PEG shell. Figure 4 indicating that CPT–SS–CPT were successfully loaded into the polymeric micelles. Moreover, these results of Figure 2 and Figure 3 indicated that the CPT–SS–CPT/PMs can efficiently be accumulated at tumor site by the enhanced permeability and retention (EPR) effect [13]. As shown in Figure 5, this stability confirmed that the dimeric CPT could effectively inhibit the aggregation or crystallization of drugs and improve the stability of the system. In order to support this result, Figure 6 further confirms that dimeric CPT–SS–CPT had low crystallinity after the cross-linking with a cystamine.

It is well-known that disulfide bonds can be cleaved with reducing reagents in cells, such as glutathione (GSH) [14,15]. The content of GSH in cancer cells is higher than in normal cells [16]. A cancer cell line, i.e., HepG2 cell line, and fibroblasts, i.e., the NIH3T3 cell line, were chosen for the cell experiment. As shown in Figure 7, with the increasing concentration of samples, cell viability decreased in both cell lines. Because the disulfide bonds in prodrugs could be triggered by GSH [17,18], the dimeric prodrug can be rapidly detached to CPT molecules, and the active CPT can kill cancer cells effectively. Moreover, the micelles of CPT–SS–CPT/PMs showed the same cytotoxicity against cancer cells as the relevant prodrugs, indicative of the ignorable influence of the amphiphilic shell on the cytotoxicity of the CPT–SS–CPT. Direct visualization of cell viability was observed using fluorescein diacetate–propidium iodinate (FDA–PI) staining. FDA, a non-fluorescent molecule, can hydrolyze by nonspecific esterases in viable cells to produce green fluorescence in the cytoplasm. PI, a nucleic acid-binding dye producing red fluorescence, cannot penetrate the membrane of viable cells, but it can readily enter the apoptotic/dead cells due to the loss of membrane integrity. Therefore, FDA-PI staining is able to distinguish viable cells (in green) and dead cells (in red) [19]. In Figure 7b, the cells treated with drugs with disulfide bonds (CPT–SS–CPT) demonstrated significant amount of red fluorescent cells. This shows that CPT–SS–CPT/PMs have a certain therapeutic effect.

## 4. Materials and Methods

Materials. Methanol, toluene, dichloromethane (DCM), chloroform (CH_3_Cl_3_), sodium hydroxide (NaOH) and tetrahydrofuran (THF) were purchased from Hengxing Chemical Reagent Co., Ltd. (Tianjin, China). Poly(ethylene glycol) (PEG, Mn = 2000 g/mol) and cystamine dihydrochloride were obtained from Shanghai McLean Biological Reagent Co., Ltd. (Shanghai, China). Triphosgene, 4-Dimethylaminopyridine (DMAP), N,N-diisopropyl-ethylamin (DIEA), stannous octoate (Sn(Oct)_2_), ε-caprolactone (ε-CL), camptothecin (CPT) were purchased from Aladdin (Shanghai, China). All reagents and solvents were received without purification. 3-(4,5-dimethylthiazol-2yl)-2,5-diphenyl tetrazolium bromide (MTT), fluorescein diacetate (FDA, >98%), propidium iodinate (PI, >98%), and d-mannitol (>99%) were obtained from Sigma-Aldrich Chemical (St. Louis, MO. USA). HepG2 and NIH3T3 cell lines were purchased from the American Type Culture Collection (ATCC, Rockville, MD, USA).

The dimeric camptothecin was synthesized through two steps: (i) To obtain purified cystamine, 4.5 g of cystamine dihydrochloride and 1.6 g of NaOH were dissolved in 50 mL of THF and stirred for 3 days at room temperature (RT). After the reaction, the solution removed the by-products through the filter paper which pore size is 15~20 μm, and then the THF is completely evaporated to obtain the final product. (ii) 30 mg of triphosgene and 122 mg of DMAP were dissolved in 10 mL CH3Cl3 and stirred for 30 min at RT under nitrogen (N2). Subsequently, the mixture solution of CH3Cl3 and THF (*v*/*v* 1:1, 10 mL) containing cystamine (36 mg) and DIEA (122 mg) was dropwise added into above solution, and continuously reacted for 3 h. Finally, 100 mL of CH3Cl3 containing 100 mg of camptothecin was poured into previous mixture solution and stirred for 12 h at the same condition. The crude CPT–SS–CPT was separated and purified by a silica gel column. Transmission electron microscopy (TEM) measurements were conducted on a JEM-1200 microscope at 120 kV. UV-vis absorbance was assayed on a SEV 500. XRD diffraction analyses were characterized by x-ray polycrystalline diffractometer (XRD) (Phillips X’pert Pro Super, The Netherlands).

PCL–PEG–PCL was synthesized via ring-opening polymerization of caprolactone (CL) using PEG as a macroinitiator. Typically, 2 g of PEG was dissolved in 25 mL anhydrous toluene and stirred for 1 h at 84 °C. Then, 2 g of CL and 5 μL of Sn(Oct)2 were added to the above solution and reacted for 12 h at 120 °C under N2. The resulting polymers were purified via dialysis with deionized water and followed by lyophilization. The molecular weight distribution of the prepared PCL–PEG–PCL three-block copolymer was 5000–7000, wherein the ratio of the molecular weight of the PCL segment to the molecular weight of the PEG segment is 1.5–3.

CPT–SS–CPT-loaded PCL–PEG–PCL PMs (CPT–SS–CPT/PMs) were prepared by mixing 300 mg PCL–PEG–PCL and 30 mg CPT–SS–CPT in 3 mL DCM and sonicated for 10 min. Then, DCM was thoroughly removed by rotary evaporation and an oil pump. A CPT–SS–CPT/PCL–PEG–PCL thin film was formed and re-dispersed in 10 mL of PBS solution to obtain the CPT–SS–CPT-loaded PCL–PEG–PCL PMs. The CPT were encapsulated into PCL–PEG–PCL PMs using the same process.

CPT–SS–CPT loading efficiency and loading content were studied and analyzed using high-performance liquid chromatography (HPLC, C18 column, 150 mm × 4.6 mm). Briefly, CPT was first dispersed in methanol and diluted different concentrations to obtain the standard curve of CPT by the peak integral of the chromatogram (as shown in Figure 2a). 10 mg of CPT–SS–CPT/PMs were dissolved in methanol (10 mL) containing 10 mg DTT to measure loading efficiency and loading content by comparing the peak integral values of the samples at 265 nm to a calibration curve of CPT concentrations.

Using the MTT assay, the cytotoxicity of the modified CPT and CPT-loaded PMs was assessed. Cultured HepG2 and NIH3T3 in Dulbecco’s Modified Eagle Medium (DMEM). Seeded cells in a 96-well plate at a density of 1 × 10^4^ cells/well. Then, incubated cells in 100 μL of DMEM/well for 24 h. The 10 μL of CPT, CPT–SS–CPT and varying concentrations of CPT–SS–CPT/PM fresh medium containing replaced the previous medium. After 24 h, 10 μL of sterile-filtered MTT reagent was added to each liver cancer cells in PBS (5 mg / ml) to a final concentration of 0.5 mg / mL. In addition, the sterile MTT reagent was filtered by microporous membrane which pore size is 220 μm. After 4 h, removed the medium and 100 μL of DMSO was added to dissolve the formed formazan crystals. Measured the absorbance at 570 nm by enzyme-labeled instrument. The cell viability (%), relative to that of control cells cultured in media without polymers, was calculated from [A]_test_/[A]_control_ × 100%, where [A]_test_ and [A]_control_ are the absorbance values of the wells (with the polyplex) and control wells (without the polymers), respectively. For each sample, ultimately the average absorbance is parallel measuring 6 wells.

Different samples (CPT, CPT–SS–CPT and CPT–SS–CPT/PMs) were treated with FDA-PI staining visually showed the survival rate of HepG2 cells. Then, HepG2 cells were seeded in 24-well plates at a density of 5×10^4^ cells/well and incubated in 500 μL of DMEM for 24 h. Then, the culture media was replaced with fresh one containing samples (0.5 μL/mL, CPT concentration). In addition, cells were incubated in the cell culture incubator for 4 h. Finally, each pore cell was stained with 8 μL PI (2 mg/mL d-mannitol) and 10 μL FDA (5 ug/mL d-mannitol) in darkness, which was imaged by a Leica DMIL Fluorescence Microscope.

## 5. Conclusions

Although CPT exhibits the highest cytotoxicity in in vitro cell experiments, it may have the best effect in future in vivo applications, or in clinical applications, in micelle-encapsulated drugs. Because of small molecule drugs, micelle-encapsulated CPT easily eliminates drug resistance, systemic toxicity, etc., including dimeric camptothecin. However, micelles do not affect the drug’s own medicinal properties, then the enhanced permeability and retention (EPR) effect reduces systemic toxicity and kill cancer cells after release from the lesion site effectively. The low crystallinity of CPT–SS–CPT was successfully synthesized using a cyatamine as a cross-linker and subsequently loaded to PCL–PEG–PCL PMs by a solvent-evaporate technique. The CPT–SS–CPT/PMs presented excellent solubility, narrow size distributions, high drug-loading efficacy, high drug-loading content and obvious cytotoxicity to HepG2 cells, which make it an advantageous candidate for improved anticancer efficacy.

## Figures and Tables

**Figure 1 ijms-20-01961-f001:**
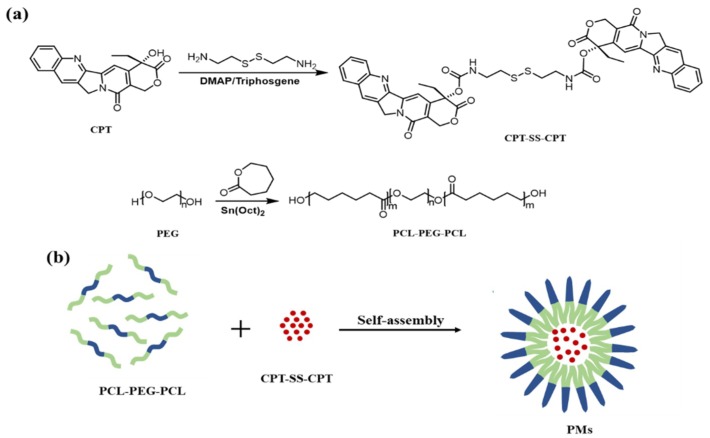
Synthesis route of a dimeric camptothecin derivative (CPT–SS–CPT) and poly(ε-caprolactone)–poly(ethylene glycol)–poly(ε-caprolactone) (PCL–PEG–PCL) triblock copolymer (**a**) and process of the self-assembly of the PCL–PEG–PCL and CPT–SS–CPT (**b**).

**Figure 2 ijms-20-01961-f002:**
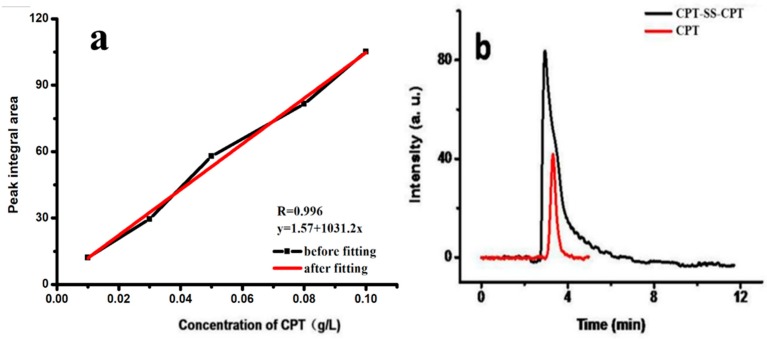
Calibration curve of CPT concentrations (**a**) and the liquid chromatogram of CPT-loaded PCL–PEG–PCL PMs and CPT–SS–CPT/PMs (**b**).

**Figure 3 ijms-20-01961-f003:**
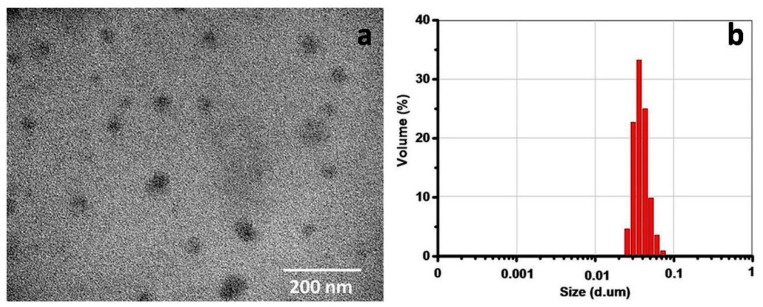
Transmission electron microscope (TEM) images (**a**) and size distribution (**b**) of CPT–SS–CPT/PMs.

**Figure 4 ijms-20-01961-f004:**
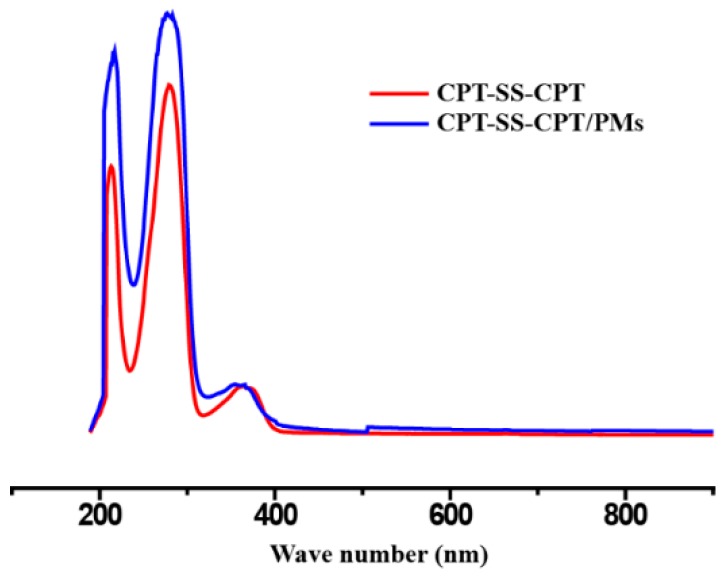
Ultraviolet-visible (UV-vis) absorption spectrum of CPT–SS–CPT in solvent and the CPT–SS–CPT/PMs in aqueous solution.

**Figure 5 ijms-20-01961-f005:**
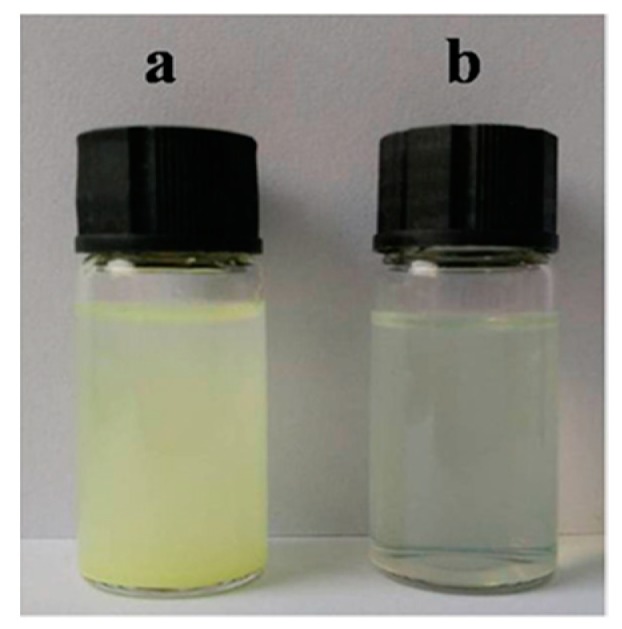
Photograph of CPT-loaded PCL–PEG–PCL PMs (**a**) and CPT–SS–CPT-loaded PCL–PEG–PCL PMs (**b**).

**Figure 6 ijms-20-01961-f006:**
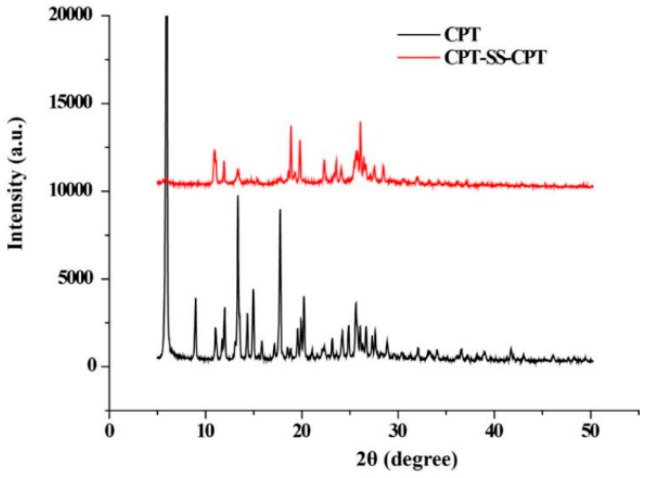
X-ray diffraction (XRD) spectra of CPT and CPT–SS–CPT.

**Figure 7 ijms-20-01961-f007:**
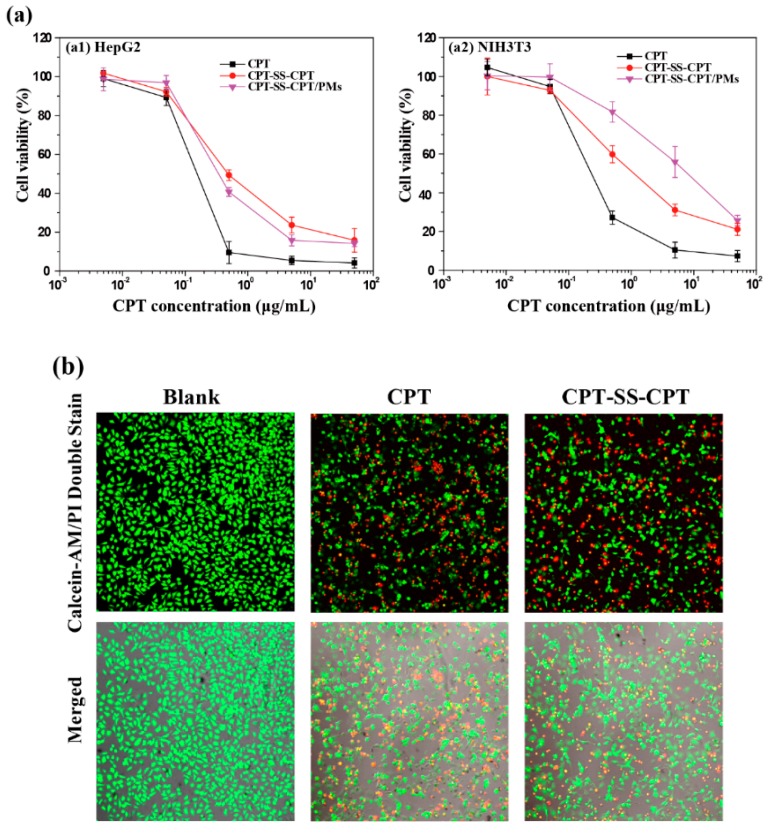
(**a**) Cell viability of the HepG2 and NIH3T3 cell lines treated with various concentration of CPT, CPT–SS–CPT, and CPT–SS–CPT/PMs (mean ± standard deviation (SD), *n* = 6,). Each data point in the graph consists of the mean and standard deviation, repeated 6 times. (**b**) fluorescein diacetate–propidium iodinate (FDA–PI) staining mediated by CPT, and CPT–SS–CPT/PMs at 0.5 μL/mL (CPT concentration; Live cells: green; dead cells: red; scale bar: 100 μm).

**Table 1 ijms-20-01961-t001:** Loading efficiency and loading content of CPT-loaded PCL–PEG–PCL polymeric micelles (PMs) and CPT–SS–CPT-loaded PCL–PEG–PCL PMs.

Sample	Drug Loading Efficiency (%)	Drug Loading Content (%)
**CPT/PMs**	9.50	0.94
**CPT–SS–CPT/PMs**	63.33	5.95

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
