# Peer review of "Construction of Dimeric Drug-Loaded Polymeric Micelles with High Loading Efficiency for Cancer Therapy"

_ijms, 2019, doi:10.3390/ijms20081961_

Round 1
Reviewer 1 Report
Authors indeed have put some efforts in the revision. However, it still hasn't met the standard to be published.
Please edit your language carefully again. There are many places with wrong English grammar even in your revision. For example, in line 175, "Measured the absorbance at 570 nm." misses the subject. in line 183, "cell culture 4 h." is not properly expressed. Again, authors must be responsible and honest with the revision.
Also, authors didn't reply to my question well. How did you filter your sample? You might have used filter paper or even syringe filter for better purification purpose. And what was the pore size?
Again, author didn't answer correctly how you took fluorescence images? For example, did you use confocal or just fluorescence microscope? If it is confocal, what is the laser wavelength? If it is fluorescence, did you use filter or laser? What is the wavelength?
Author Response
Comments: Please edit your language carefully again. There are many places with wrong English grammar even in your revision. For example, in line 175, "Measured the absorbance at 570 nm." misses the subject. in line 183, "cell culture 4 h." is not properly expressed. Again, authors must be responsible and honest with the revision.
Response: As suggested, we have edited language carefully again. (Line 181, 4th Paragraph, Page 7; Lines 189-190, 4th Paragraph, Page 7)
Comments: Authors didn't reply to my question well. How did you filter your sample? You might have used filter paper or even syringe filter for better purification purpose. And what was the pore size?
Response: We added relevant experimental details in article. (Lines 143-144, 4th Paragraph, Page 6; Lines 178-179, 4th Paragraph, Page 7)
Comments: Again, author didn't answer correctly how you took fluorescence images? For example, did you use confocal or just fluorescence microscope? If it is confocal, what is the laser wavelength? If it is fluorescence, did you use filter or laser? What is the wavelength?
Response: Thanks for your suggestion, figure 5 is just a normal photograph and not a fluorescent image. The photograph is only used to show that the stability of CPT-SS-CPT/PMs is better. We had corrected this error. (Line 59, 2rd Paragraph, Page 2; Line 88, figure 5, Page 4)
Reviewer 2 Report
Article looks much better now and can be published after polishing text.
Author Response
As suggested, the language and description were polished by a native speaker.
Reviewer 3 Report
The presented work has a clear practical orientation - targeted delivery of drugs to a living organism, that is up-to-date topic.
However, a noticeable flaw in this work is the lack of a characteristics of a three-block copolymer synthezided by authors : the total molecular mass, composition of copolymers, length of the terminal chains in comparison of central one.
I wish that this information will be given in this manuscript.
Author Response
Comments: However, a noticeable flaw in this work is the lack of a characteristics of a three-block copolymer synthezided by authors : the total molecular mass, composition of copolymers, length of the terminal chains in comparison of central one. I wish that this information will be given in this manuscript. Response: Thanks for your suggestion, We have added the description of the relevant characterization of PCL-PEG-PCL three-block copolymer. (Lines 158-160, 4th Paragraph, Page 6)
This manuscript is a resubmission of an earlier submission. The following is a list of the peer review reports and author responses from that submission.
Round 1
Reviewer 1 Report
The manuscript introduced a polymeric micelles system with the help of camptothecin to enhance the drug delivery. The structure of the manuscript is good. However, there are several serious problems of the article.
The article is too short.
There are many details missing. Please check throughout the whole manuscript. For example, how did you filter your sample. How did you take the fluorescence image...
There exist many problem with the language. You omit the subject in front of verbs more than one times.
The English style needs to be perfected. You need to focus on the unit expressions.
From both Figure 7a and b, I don't see the advantages of the new drug delivery system.
Reviewer 2 Report
This article may have a high impact for anticancer investigators, because polymeric micelles (PMs) have widely been applied for cancer treatment.
The authors reported that the disulfide-linked camptothecin (CPT) was effectively embedded in the core of the poly(ε-18 caprolactone)-poly(ethylene glycol)-poly(ε-caprolactone) (PCL-PEG-PCL) PMs for improving drug loading efficiency, where PEG can be used as a hydrophilic shell. The dimeric CPT-loaded PCL-PEG-PCL PMs exhibited excellent solubility in PBS and showed cytotoxicity for cancer cells.
The authors developed polymeric micelles based on poly(ε-caprolactone)-poly(ethylene glycol)-poly(ε-caprolactone) (PCL-PEG-PCL) triblock copolymer for efficient delivery of CPT. However, a biocompatibility and biodegradability of the new composed drug, were neither demonstrated, nor analyzed from literature.
The synthesis, characterization and description of physical–chemistry properties of the novel MCs are presented, as well as loading efficiency and content of CPT-loaded PCL-PEG-PCL PMs and CPT-SS-CPT-loaded PCL-PEG-PCL PMs were described too.
Below there are some remarks:
Line 53. The stability of CPT-loaded PCL-PEG-PCL PMs and CPT-SS-CPT/PMs was evaluated using fluorescence imaging. However, the authors proposed novel copolymer for anticancer treatment and would be nice to see results on the copolymer properties: stability of micelles at different pH, as well as osmolarity, and preincubation with BSA.
Line 130. “All reagents and solvents were received without purification.” Which reagents were purified?
Line 138. “After the reaction, the solution was filtered to remove the by-products, and then the solvent is completely evaporated to obtain the final product.” Please, describe this process.
Line 147 Sentence “XRD diffraction analyses were characterized by x-ray diffraction (XRD)” needs to be rewrite.
Line 155 “Then, DCM was thoroughly removed by rotary evaporation.” What conditions were used here?
Line 166. “Using the MTT assay assess the cytotoxicity of the modified CPT and CPT-loaded PMs. Sorry, but what reference wavelength was used? How the final data was calculated?
I would like to recommend for improving of the cytotoxicity section to include in it description of differences between captothecin, dimeric camptothecin (CPT-SS-CPT) and PCL-PEG-PCL PMs.
Reviewer 3 Report
The problem of drug delivery to the desired target in a living organism is a popular topic in which various delivery systems, including those based on polymeric carriers, are considered and studied. This work is devoted to the study of the hydrophobic anticancer drug - camptothecin, its modification, in order to prevent unwanted crystallization, and possible delivery to a living organism using poly (ε-caprolactone) -poly (ethylene glycol) -poly (ε-caprolactone) micelles as a drug delivery system.
The main point which must be clarified concerns the absence of any molecular characteristics of block copolymers synthesized and used in the work. The composition and molecular masses of the PCL-PEG-PCL samples should be listed. The latter are especially important when using such drug containing polymer systems to introduce in a living organism.
In addition, I will make some additional comments.
1. In Figure 1, poly (ε-caprolactone) macromolecules are presented in the form of rigid rods. The drawings are beautiful, but the conformation of the rod is hardly suitable for poly (ε-caprolactone) macromolecules.
2. Fig.2 It is advisable to draw a calibration straight line through the experimental points using the least squares method. I also expect in this case that the linear correlation coefficient will be r> 0.9.
3. Fig.7 What does mean “± SD, n = 6, * p<0.05”. Give, please, the explanation.